# Physics-Informed Graph Convolutional Network for Data-Free Learning of Channel Flow Fields

## Abstract

Physics-informed neural networks (PINNs) have gained traction for solving partial differential equations (PDEs) by embedding physical laws directly into the learning objective. However, traditional PINNs, typically built on fully connected neural networks, often struggle to capture localized interactions, scale poorly with mesh complexity, exhibit limited generalization across geometric variations and suffer from training instability due to minimal spatial representation. To address these challenges, we propose physics-informed graph convolutional network (PIGCN) architecture that leverages mesh connectivity to enforce local spatial coupling between variables, enhancing the ability of the model to learn structured physical relationships. In addition, this work investigated on data-free learning to reduce reliance on data observation and improve the generalization capability, introduced global physics loss, and implemented two-level dynamic weighting scheme to adaptively balance between multiple PDE residuals (e.g., continuity, momentum, energy) and composite loss terms (e.g., boundary conditions, global physics, and data), improving convergence in multi-objective training. We evaluated PIGCN against PINN on a square channel with isothermal walls and L-shaped channel with bending flow, which exhibits complex fluid dynamics phenomena. For square channel geometry, baseline PIGCN outperformed baseline PINN by 56.6%, and was further improved with two-level dynamic weighting by 38.9%, which achieved 66.6% in rMSE error improvement compared to equivalent PINN. In L-shaped channel, baseline PIGCN improved rMSE error by 35.7% against baseline PINN, and the introduction of global physics loss further reduced rMSE error by up to 71.3% in comparison with baseline PIGCN. Our investigation also demonstrated that PIGCN achieved significantly faster training time and less graph memory consumption than PINN on various geometries, by 35.8% and 62.7% respectively. These results validated that graph architectures and hierarchical loss weighting can substantially enhance performance of physics-informed machine learning models for fluid dynamic analysis with scalable resource usage.

## 1 Introduction

As machine learning (ML) emerges into scientific computing, physics-informed neural networks (PINNs) created a breakthrough by embedding governing physics directly into the loss function of a neural network to maintain the compliance with physics laws which addresses the limitation of conventional data-driven learning. PINNs can achieve accurate predictions while leveraging known physics to compensate for scarce or even no data, and they can also learn parametric solution families in one training (Cho et al., 2024). This concept builds upon earlier attempts to solve PDEs with neural nets in the 1990s (Dissanayake & Phan-Thien, 1994; Lagaris et al., 1998) but was popularized by Raissi et al. (2019) who demonstrated that neural networks constrained by physical laws can solve forward and inverse problems. Notably, Rao et al. (2020) and Jin et al. (2021) applied PINN to simulate laminar and turbulence incompressible flows, Cai et al. (2021) extended the application to thermal-fluid problems, Hao et al. (2024) created a comprehensive benchmark for PINN. Despite the strengths, PINNs yet face significant challenges when applied to complex problems, especially without any simulated or empirical data.

The major limitation PINNs is limited capability to capture local interactions due to the lack of spatial information between disconnected points (Rathore et al., 2024). Recent studies have reported that PINNs often struggle to achieve the same accuracy as classical CFD in such regions, requiring very deep networks or excessive training time (Wang et al., 2022; Eivazi et al., 2022).

Many practical problems involve irregular geometries or unstructured meshes which requires more flexibility. Graph neural networks (GNNs) can represent the computational mesh or point cloud as a graph (Gori et al., 2005; Scarselli et al., 2009) and already demonstrated its potential as PDE solver in data-driven fluid mechanics. For example, Sanchez-Gonzalez et al. (2020); Pfaff et al. (2021) simulated vortex shedding or airflow around objects with good accuracy. To leverag GNNs within physics-informed learning (PIGNNs), message-passing architecture is used to propagate information locally, while the loss term penalizes violations of the governing equations. Early studies report that PIGNN can speed up convergence and improve solution accuracy for fluid flow problems on unstructured grids (Gao et al., 2022). This success has inspired PIGNN to further reduce reliance on data-driven learning compared to data-driven GNN methods (Gao et al., 2022; Peng et al., 2023; 2024). However, most PIGNN works were conducted on multi-layer perceptron (MLP)-based message-passing blocks, while only few preliminary works discussed the potential of graph convolutional network (GCN) despite its superior ability of learning local information by mixing a node feature with those of its neighbors. Peng et al. (2023) implemented a framework to incorporate physics constraints to GCN, but its heavy reliance on full field data observation limited its generalization capability. In addition, the accuracy of the proposed model were improved slightly compared to pure data-driven model which only cost about 25% training time. A recent work by Zhang et al. (2025) also demonstrated a GCN-based surrogate model with satisfactory accuracy, but on less complex PDEs, which only involves single equation or first-order derivatives, predicting only one or two field variables. Hence the potential of physics-informed GCN on high-dimensional, high-order PDEs with limited data or no data remains underexplored.

While architecture innovation is a major breakthrough for high-fidelity multiphysics prediction, some challenges observed in PINN will remain to be tackled. In multiphysics problems, field values (e.g., pressure, velocities, temperature) often vary in different scales, typically in orders of magnitude. In result, the physics loss terms (e.g., continuity, momentum, energy) can be poorly balanced, potentially leading to vanishing or exploding gradient issues (Xiang et al., 2022; Perez et al., 2023). For example, standard PINN may fail to enforce the divergence-free condition or capture sharp thermal gradients without loss weights being carefully tuned (Li & Feng, 2022; Schwencke & Furtlehner, 2025; Song et al., 2024).

In this work, we explore a Physics-Informed Graph Convolutional Network (PIGCN) for predicting two-dimensional (2D) incompressible fluid flow without any data. We compare PIGCN against conventional PINN as baseline to evaluate the benefits of the GCN architecture. Furthermore, we investigate the impact of two training enhancements: dynamic loss weighting on both PDE residual terms and composite loss terms and addition of global physics constraint. The proposed model is tested on thermal-fluid and fluid dynamics problems to assess its ability to handle different levels of system complexity. In summary, our contributions include: (1) formulating a GCN to physics-informed learning for data-free learning, demonstrating the ability of PIGCN to handle high-dimensional, high-order PDEs; (2) demonstrating that adaptive loss weighting further improve convergence more effectively in PIGCN compared to PINN; (3) introducing a global flux constraint term which significantly improved prediction accuracy in complex phenomena. The following sections present our problem definition, methodology, experimental results and findings in detail.

## 2 PROBLEM FORMULATION

### 2.1 PROBLEMS

To evaluate our PIGCN approach, we are inspired by the heat sink designs in electronics cooling and tested on different geometric or system complexities to test the robustness of the model. We considered two representative 2D channel configurations: a straight rectangular channel with heated wall and an L-shaped channel. These configurations represent commonly used cooling designs - from simple straight thermal-fluidics passages to sharp turns that disrupt flow paths. The straight channel offers a baseline scenario with fully developed flow, and the L-shaped channel introduces a sudden change in flow direction that can cause separation and recirculation, making the system

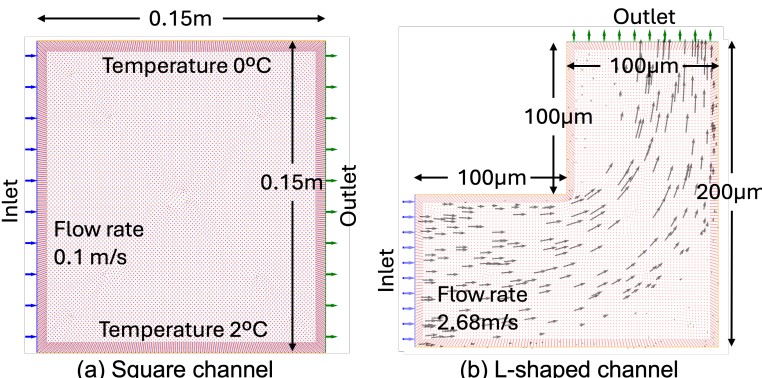

Figure 1: An illustration figure of the geometries being evaluated in this work. (a) square duct; (b) L-shape channel

more complex and unpredictable. By investigating these cases, we simulate simplified yet extendable channel scenarios, benchmarking the model performance under conditions analogous to those encountered in actual electronic cooling.

**RECTANGLE** The 2D rectangular channel problem represents a straight line coolant passage, with length and height of 0.15m each as illustrated in Fig. 1 (a). Air flow is injected with 0.1m/s at the channel inlet on the left boundary and exits at the outlet on the right boundary where pressure is enforced at 0 Pa. The bottom wall is heated at 275K while the top wall is maintained at 273K, which introduces thermal gradients into the system and couples fluid flow with heat transfer. Both top and bottom boundaries are assumed to be no-slip walls. The thermofluidic problem predict four field variables (pressure, temperature, x-velocity, and y-velocity).

**L-SHAPE** The L-shaped channel introduces a sharp 90° turn in the flow path, with length and height of 200$\mu m$ each as depicted in Fig. 1 (b). Water flow is injected with 2.68m/s at the channel inlet on the left boundary with 100 $\mu m$ opening, bends in sharp corner and exits at the outlet with 100 $\mu m$ on the top boundary where pressure is enforced at 0 Pa. All other boundaries are assumed to be no-slip walls. Without any temperature gradient, the fluid dynamics problem predict three field variables (pressure, x-velocity, and y-velocity). The L-SHAPE case serves as a realistic challenge, ensuring that PIGCN can handle flow re-direction and the associated localized phenomena common in microchannel cooling networks.

## 2.2 NON-DIMENSIONALIZED NAVIER-STOKES EQUATIONS

The general governing equations for fluid systems are the conservation of mass, momentum, and energy equations. For physics-informed machine learning models, non-dimensionalization of the field variables is necessary to maintain stable gradients as solving PDEs often requires high-order derivatives. The details of non-dimensionalization process is available in Appendix A.1. Note that energy equation only applies when temperature profile presents (thermal-fluid problems), which means for fluid dynamic problems without thermal factor, only other three equations will be considered.

## 3 METHODOLOGY

### 3.1 PHYSICS-INFORMED GRAPH CONVOLUTIONAL NETWORK

Our PIGCN model leverages graph convolution operations to learn the solution fields on a mesh, explicitly respecting the spatial relationships defined by the physical domain in addition to the physical constraints. In GCN, each node (i.e., mesh point) updates its state by aggregating information from neighboring nodes connected by the edge. This process is analogous to a standard convolution in image processing, except that the convolution kernel is applied over an irregular neighborhood defined by the mesh connectivity instead of a fixed grid. By passing messages along graph edges,

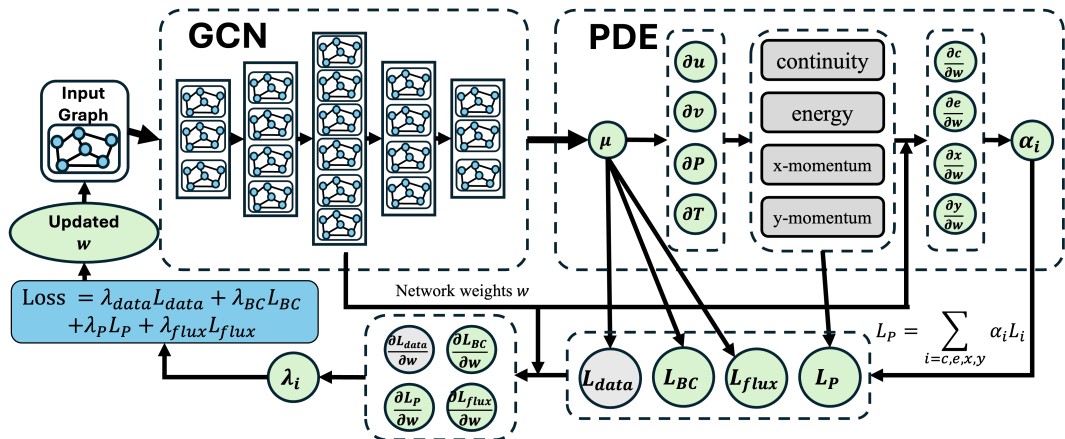

Figure 2: The architecture of physics-informed graph convolutional network (PIGCN). Note data loss with in light gray circle is optional.

the model naturally captures local spatial couplings: each node features are influenced directly to its immediate neighbors, mirroring the localized interactions of physical laws (e.g., heat diffusion or fluid momentum exchange occurring between adjacent regions).

Graph convolutions are particularly well-suited for mesh-based physical problems as they incorporate a structured inductive bias about locality and geometry. The network's weight-sharing mechanism provides the ability that a single learned rule from one neighborhood is reused across the domain, matching the characteristics of physical laws having uniform forms. Moreover, PIGCN does not require a rectangular grid and can directly handle arbitrary shapes and non-uniform discretizations. This approach significantly reduces the number of parameters and ensures that the model respects domain connectivity, which improves learning efficiency and solution accuracy on real-world multiphysics problems.

To incorporate mesh connectivity, we construct an undirected graph with nodes on the collocation and boundary points. Each node carries its coordinates, while each edge carries a scalar distance feature. Then, we apply a stack of graph-convolution layers (Kipf & Welling, 2017). At each step, a node updates its embedding by aggregating transformed messages from its neighbors. A per-node linear head predicts the field values $(U, V, P, T)$, x- and y-velocities, pressure and temperature, respectively. The key advantage of PIGCN is that it learns and enforces the local coupling of fields imposed by the mesh geometry, yielding sharper resolution of boundary layers and corners than the baseline PINN counterpart. Figure 2 illustrates the framework of our proposed model: the network takes the mesh-based graph as input, progresses through multiple GCN layers, each followed by a nonlinear activation layer, then forms the prediction for all fields values with a linear layer. Boundary condition loss (and optionally data loss) is directly computed by comparing the prediction with reference values, while physics residuals are calculated from the mismatch between PDE and predictions.

## 3.2 Loss formulation

PIGCN seek a function $u(x; \mathbf{w})$ ($\mathbf{w}$ denotes network weights) such that the following conditions are met simultaneously:

- minimizes deviation from simulated or empirical data,

- satisfies prescribed boundary conditions,

- drives the governing PDE residuals to zero.

Consequently, the loss components are:

$$L_{data} = \frac{1}{N_d} \sum_{i=1}^{N_d} \left\| u(x_i^d; \mathbf{w}) - u_i^{true} \right\|^2,$$

$$L_{bc} = \frac{1}{N_{bc}} \sum_{i=1}^{N_{bc}} \left\| B[u(x_i^{bc}; \mathbf{w})] \right\|^2, \quad L_P = \frac{1}{N} \sum_{i=1}^{N} \left\| D[u(x_i^p; \mathbf{w})] \right\|^2 \tag{1}$$

where $D[\cdot]$ encodes the differential operator of the PDE and $B[\cdot]$ enforces boundary conditions, $u^{true}$ is observation from dataset, $x^d, x^{bc}$ and $x^p$ represent data points, boundary points and interior collocation points respectively. Similarly, $N_d, N_{bc}$ and $N_P$ stand for number of corresponding points.

The composite loss function is $L = L_{data} + L_{bc} + L_P$. Note that $L_{data}$ only applies to data-assisted training scenarios, and is not included for data-free training. The physics residual $L_P$ decomposes into four coupled PDE equations described in Eq. 3 in Appendix: $L_P = L_c + L_u + L_v + L_e$, whereas the losses are computed over a set of interior collocation points on continuity, x-momentum, y-momentum and energy equations, respectively.

In addition to the classical loss terms, we augment the loss function with a flux conservation term $L_{flux}$ to enforce global physical laws. It leads to notable improvement in the L-shape design with bending flow, where major flow changes in a different axial direction. This flux loss is designed to ensure conservation of mass by penalizing any discrepancy between inflow and outflow through the domain. It effectively guides the model to avoid non-physical solutions that might satisfy local PDE residuals but violate global flux conservation, especially in the absence of data. Accordingly, our total loss becomes: $L_{\text{total}} = L_{data} + L_{bc} + L_P + L_{flux}$.

### 3.3 Adaptive loss weight balancing

Based on the fact that these four residuals can differ in scale by orders of magnitude, a uniformly unweighted sum often leads to under-optimization of the smaller-scale equation or over-fitting of the larger. Therefore, we explore three distinct weighting strategies: static weighting, PDE dynamic weighting, and two-level dynamic weighting.

Previous works have discussed higher data loss weight could improve the performance of PINN (Barreau & Shen, 2025), however, tuning loss term weights between different PDE loss terms has not been much discussed. First, we prescribe constant coefficients to each term. It is to tune the PDE loss weights of terms with different scale as hyperparameters, so that they result in roughly same order of magnitude.

Next, applying the GradNorm algorithm proposed by Chen et al. (2017), we introduce four trainable weights $\{\alpha_c, \alpha_u, \alpha_v, \alpha_e\}$ to the PDE residual losses and jointly optimize them with $\mathbf{w}$, to equalize the norm of each PDE residual gradient. At each training iteration, we measure the gradient norm of each weighted loss, compute their average, and define the relative inverse training rate for each loss as its value divided by the mean loss. Then, we set a target gradient magnitude for each term by scaling the average gradient by this relative rate raised to a tunable power $\gamma > 0$. The weights are updated by minimizing the absolute difference between the current and target gradient norms, which makes the gradients of the weighted residuals track their targets and balances convergence.

To capture the hierarchical structure of our objectives, we apply GradNorm at two levels: first adapt $\{\alpha_c, \alpha_u, \alpha_v, \alpha_e\}$, to ensure all four physics-residual gradients share a similar magnitude. This yields a composite $L_P = \sum_i \alpha_i L_i$. Next, considering $L_{data}, L_{bc}, L_P$ and $L_{flux}$ as composite losses, we apply GradNorm again and introduce $\lambda = (\lambda_{data}, \lambda_{bc}, \lambda_P\ \lambda_{flux})$ to balance physics enforcement, boundary condition compliance, and observance data. Thus, at epoch $t$, the full trainable loss becomes:

$$L(\mathbf{w}) = \lambda_{data}(t)L_{data} + \lambda_{bc}(t)L_{bc} + \lambda_{flux}(t)L_{flux} + \lambda_P(t) \sum_{i \in \{c,u,v,e\}} \alpha_i(t)L_i \tag{2}$$

with both sets of weights updated online to equalize their respective gradient norms.

## 4 EXPERIMENTS AND RESULTS

### 4.1 EXPERIMENT SETUP

We conducted experiment on a 2D steady-state fluid flow in two geometries, using high-fidelity CFD solutions for references. Since GCN does not require regular mesh to be trained on, we were able to insert inflation layers at each boundary of the domain to ensure the high-gradient information on the corners were well-captured. We considered both data-assisted training - where ground truth field values are included in the loss - and data-free training using only PDE residuals, boundary and flux constraints. For evaluation, we first constructed a evenly-spaced finer graph in the domain, and compared the predicted pressure $P$, velocity components $U, V$, and temperature $T$ (in rectangle geometry only) against the high-fidelity CFD reference via visual field plots and relative mean squared error (rMSE). All network and training details were given in Appendix A.2 and A.3.

### 4.2 PIGCN VS. PINN IN RECTANGLE DOMAIN

We first implemented the baseline PIGCN without any enhancements. The networks take the spatial coordinate $(x, y)$ as input, and output the four approximated solution fields $(U, V, P, T)$. By applying PyTorch's automatic-differentiation (Paszke et al., 2019) we computed the continuity, momentum, and energy residuals at every interior point, as well as boundary condition mismatches.

As an initial baseline, we trained both networks on a square cavity with isothermal walls including training data reference from CFD simulation using Ansys Fluent. Both models achieved satisfactory prediction accuracy that closely match the CFD ground truth under these data-rich conditions. In this data-assisted scenario, the PINN and PIGCN reconstructions were qualitatively similar, each capturing the overall flow patterns and gradients, and their rMSE errors were also almost identical, at $1.40 \times 10^{-4}$ and $1.35 \times 10^{-4}$ respectively.

While both models provided satisfactory prediction accuracy, data may not be available in many real-world cases, and it can limit generalization capability, which can require retraining of the model with any design changes. Next, we compared the data-free learning performance of PINN and PIGCN on the same square convection problem, using only the governing equations and boundary conditions for training. As shown in Figure 3, although both baseline PINN and baseline PIGCN successfully recovered the key flow features (e.g., the hotter rising fluid along bottom side and the centering flow trend) purely from physics constraints, PIGCN resulted in more accurate prediction on high gradient corners in pressure and v-velocity fields. PIGCN achieved an rMSE of $1.85 \times 10^{-3}$, whereas PINN struggled at $4.27 \times 10^{-3}$. PIGCN more effectively leveraged the mesh-inductive bias to satisfy the PDE constraints across the domain, yielding solutions much closer to the high-fidelity CFD reference. This gap aligned with previous findings that while PINNs can solve simpler PDEs in data-free scenarios, they often struggle to produce reliable predictions for complex multiphysics problems – an issue mitigated by PIGCN's architectural enhancements.

To examine the effect of network design, we performed an architecture ablation on the square channel under the same data-free training protocol. The detailed results are shown in Appendix A.2. Our proposed symmetric PIGCN best matched the CFD reference across all fields. A uniformly wide 8-layer GCN (all 128 channels, following the design in Peng et al. (2023)) produced visibly over-emphasized pressure spots, weaker boundary layers, and distorts corner eddies in both velocity fields, indicating poorer fidelity. A shallower symmetric with less channels per layer variant underfitted and missed fine-scale structures, while a deeper symmetric model also yielded more inaccurate predictions, especially at bottom left corner of y-velocity field. These results suggested that a multi-scale channel schedule is more effective than a flat, uniformly wide stack for capturing coupled multiphysics features in data-free PIGCN training, and the shape of the network needs to be selected carefully as well.

### 4.3 DYNAMIC LOSS WEIGHT CONFIGURATION

Adaptive loss balancing further improved the accuracy and convergence of PIGCN on the 2D thermofluidic problem in rectangular channel. As illustrated in Fig. 4, applying adaptive weighting on PDE residuals yielded a notable improvement both visually and quantitatively, from $1.85 \times 10^{-3}$ to $1.05 \times 10^{-3}$, a 43.5% reduction. A joint-adaptive weighting of both PDE residual terms and the

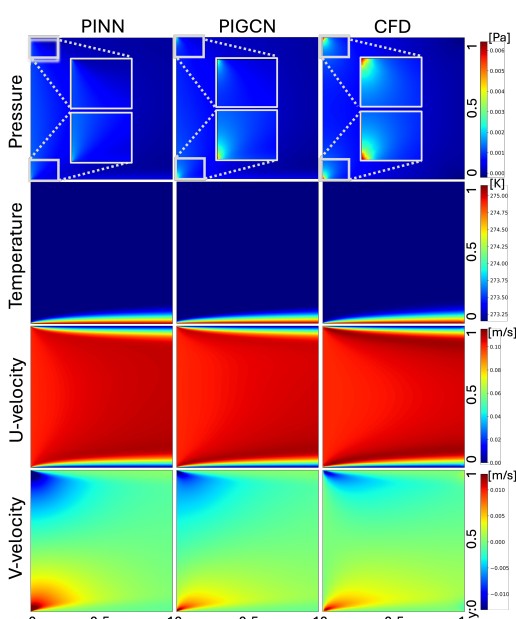

Figure 3: Comparison between predictions of data-free PIGCN and baseline PINN.

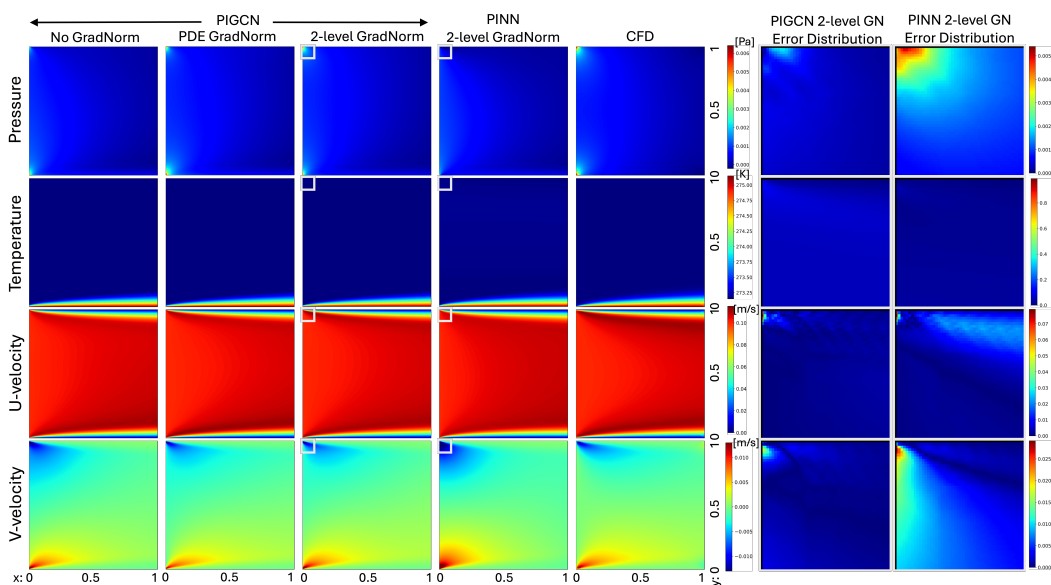

Figure 4: PIGCN predictions with single and two levels of dynamic weighting optimization and PINN with two levels of dynamic weighting. Left to right: (1) PIGCN without GradNorm; (2) PIGCN with GradNorm on PDE loss weights; (3) PIGCN with GradNorm on both PDE and composite loss weights (2-level); (4) PINN with GradNorm on both PDE and composite loss weights (2-level); (5) CFD reference; (6) error map of PIGCN with 2-level GradNorm; and (7) error map of PINN with 2-level GradNorm. Error maps are zoomed into the marked areas at top-left corner of high gradient region for better visualization.

composite losses (here, balancing between boundary constraints and physics) led to $1.13 \times 10^{-3}$ error. In comparison, the =PINN with the same two-levels weighting reached $3.38 \times 10^{-3}$ error, 20.8% improvement comparing to baseline PINN, but still about 3x higher than PIGCN. These results underscored the effectiveness of our loss-balancing scheme, which was necessary in cases

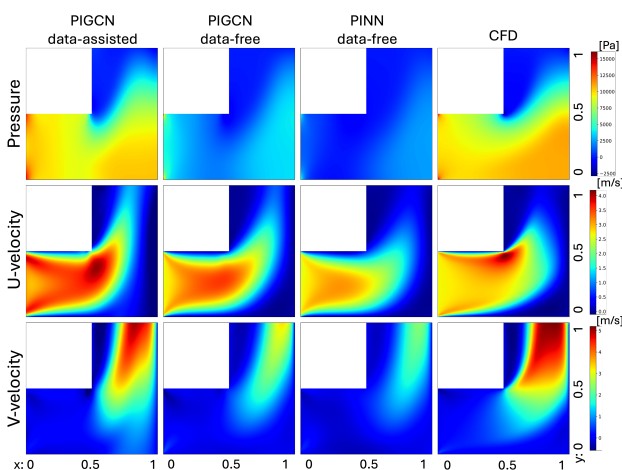

Figure 5: Data-assisted PIGCN, data-free PIGCN and PINN predictions on L-shape geometry.

similar to the square channel, as x-velocity term dominated y-velocity, causing imbalance between momentum residuals in Navier-Stokes equations.

### 4.4 L-SHAPE GEOMETRY AND GLOBAL PHYSICS LOSS

We also experimented on a more complex L-shaped channel that involves a 90° bend and recirculation zones. From illustration in Fig. 5, under full supervision, PIGCN achieved rMSE of $7.62 \times 10^{-2}$ on this domain indicating reasonable accuracy. However, without any training data, the problem became far more challenging. The baseline PINN completely under-predicted all fields, missed high-pressure regions and flow structures, converging to an rMSE of 1.262, whereas PIGCN attained 0.810. Although the prediction result was still unsatisfactory, error was quantitatively lower and visually obvious in field plots. The graph-based architecture better captured localized flow features (e.g., corner eddies, jets) in the absence of data, yielding a solution matching the CFD ground truth more closely in critical regions.

We further examined whether incorporating physical priors can boost data-free learning performance on the L-shaped design. In particular, we added a global flux-conservation loss (i.e., penalizing any imbalance between total inflow and total outflow) and studied its impact at different weightings. Figure 6 summarized the results. Even a small flux loss weight (0.01) yielded a noticeable improvement: error of PIGCN dropped from 0.810 to 0.465 (a 42.7% reduction). Increasing the weight to 0.05 reduced the error to 0.233 (71.3% lower than baseline). Interestingly, when flux loss weight was low (less than 0.1), the inlet flow was more emphasized with increased flux loss weight and predicted higher pressure and u-velocity near inlet. However, when flux loss weight further increased, the emphasis on the inlet started to diminish. This is due to the enlargement of flux loss, other loss terms were suppressed and the model tended to find the solution with least flux difference. Overall, this experiment demonstrated that PIGCN can effectively leverage partial physics knowledge to compensate for scarce data. By enforcing global conservation laws in addition to local PDE residuals, PIGCN dramatically enhanced solution quality.

### 4.5 COMPUTING RESOURCE COST COMPARISON

Finally, we compared the computational efficiency of PIGCN and PINN. Consistent with our design, the PIGCN model trained substantially faster and with lower memory usage than its fully-connected counterpart. For our rectangle geometry, we generated multiple reference meshes with different length/height ratio and element size, thus controlling the number of points available for training. In our tests summarized in Table 1, PIGCN required only around 60-70% of the PINN's wall-clock time and about 35-40% GPU memory for the same problem, while maintaining same or higher performance. The speedup arised from two factors: (i) PIGCN has 60% fewer trainable parameters than the dense network, and (ii) its localized message-passing computations scale linearly with the

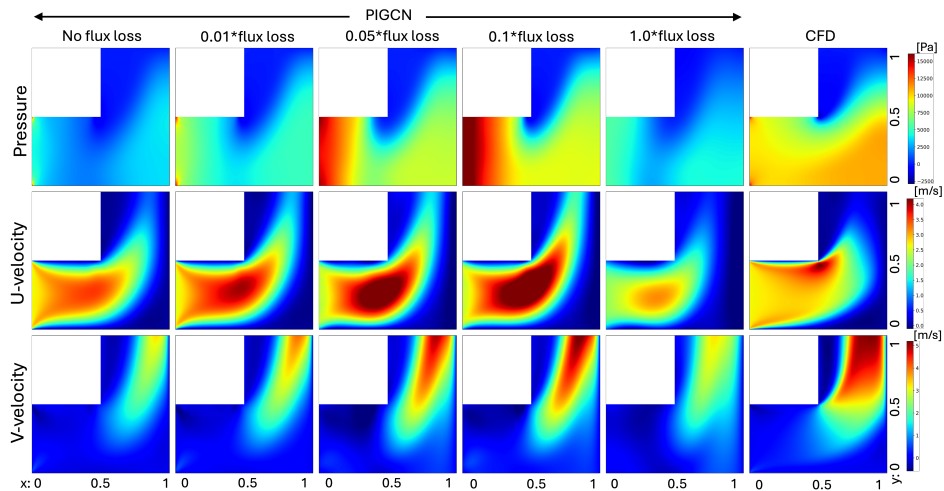

Figure 6: PIGCN predictions with different flux loss weights. Left to right: no flux loss, 0.01 flux loss weight, 0.05 flux loss weight, 0.1 flux loss weight, 1.0 flux loss weight, and CFD reference.

Table 1: Training time and graph memory usage of PINN and PIGCN on various sizes and mesh granularity of rectangular channel geometry

| Number of points | PINN | | PIGCN | | Improvement | |
|---|---|---|---|---|---|---|
| | Memory (GB) | Time (h) | Memory (GB) | Time (h) | Memory (%) | Time (%) |
| 18,571 | 9.56 | 6.5 | 3.66 | 4.5 | 61.75 | 30.77 |
| 40,861 | 21.58 | 23 | 7.53 | 14 | 65.12 | 39.13 |
| 53,843 | 28.31 | 25.7 | 10.65 | 15.5 | 62.38 | 39.69 |
| 77,065 | 39.63 | 27.9 | 15.01 | 18.1 | 62.11 | 35.13 |
| 144,529 | 74.31 | 39.5 | 28.15 | 26 | 62.12 | 34.18 |

number of edges per node (as opposed to quadratic scaling in a fully-connected layer with all points). In practice, these efficiency gain enabled PIGCN to handle finer meshes and larger domains on the same hardware budget. For instance, given the significantly smaller memory footprint, one can train PIGCN on a much denser mesh to achieve higher accuracy without running out of resources. This resource advantage, combined with the accuracy improvements noted above, highlighted the practical value of the proposed PIGCN approach for accelerating physics-informed modeling of complex flows.

## 5 CONCLUSION

We have presented a data-free Physics-Informed Graph Convolutional Network (PIGCN) framework for steady 2D incompressible fluid flow, together with a two-level dynamic loss-weighting scheme that simultaneously balances the PDE residuals and other physical and data constraints. By exploiting mesh connectivity via message-passing layers, the PIGCN captures local boundary-layer and corner phenomena more effectively than a fully connected baseline. We also introduced a global physical constraint to ensure the volumetric flow rate consists between inlet and outlet. Our extensive data-free experiments - comparing static vs. single-level vs. two-level dynamic weights and flux loss weight experiments demonstrate that the proposed approach yields significantly lower errors both visually and quantitatively compared to conventional PINN across a range of mesh densities while achieving remarkable savings on training time and memory space usage.

In future work, we will extend this PIGCN methodology to unsteady and three-dimensional flows, explore graph-based domain decomposition techniques on more complex geometries, and tackle adaptive graph refinement during training process to further improve the predicting performance.

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

## A APPENDIX

### A.1 NAVIER-STOKES EQUATION NON-DIMENSIONALIZATION

For most machine learning techniques, it is important to keep the output values at the same scale, to ensure smooth and stable optimization process and avoid ill-conditioned gradients, hence normalization or standardization is a crucial step for training pipelines. However, for physics-informed models, although scaling output is even more important, simple min-max normalization cannot be applied directly to physical values, because the predictions will be further incorporated into PDE residuals and arbitrarily modifying such values causes unalignment between prediction and underlying physical laws. To address the issue, non-dimensionalized PDEs are usually required for PINN training to ensure stable gradients of higher derivatives to calculate PDE residuals.

The continuity, x-momentum, y-momentum, and energy equations in Cartesian coordinates for 2D-steady problem can be respectively written as

$$\frac{\partial U}{\partial x} + \frac{\partial V}{\partial y} = 0$$

$$U\frac{\partial U}{\partial x} + V\frac{\partial U}{\partial y} + \frac{1}{\rho_0}\frac{\partial P}{\partial x} - \nu\left(\frac{\partial^2 U}{\partial x^2} + \frac{\partial^2 U}{\partial y^2}\right) = 0$$

$$U\frac{\partial V}{\partial x} + V\frac{\partial V}{\partial y} + \frac{1}{\rho_0}\frac{\partial P}{\partial y} - \nu\left(\frac{\partial^2 V}{\partial x^2} + \frac{\partial^2 V}{\partial y^2}\right) - \frac{\rho}{\rho_0}g = 0 \tag{3}$$

$$U\frac{\partial T}{\partial x} + V\frac{\partial T}{\partial y} - \frac{k}{\rho_0 c}\left(\frac{\partial^2 T}{\partial x^2} + \frac{\partial^2 T}{\partial y^2}\right) = 0$$

where $\nu$ is dynamic viscosity, $k$ is fluid conductivity, $c$ stands for heat capacity, and $g$ is gravitational acceleration. The characteristic variables are $U_0$ (e.g., free stream or inlet velocity), $\rho_0$ (e.g., average density), and $T_{max} - T_{min}$ (e.g., reference temperature difference). In the equation, there are four physical values: pressure $P$, temperature $T$, and normal velocity components $\mathbf{u} = (U, V)$ along with spatial coordinates $x, y$. To maintain validity of the equation, the values are non-dimensionalized such that:

$$x^* = \frac{x}{w} \quad y^* = \frac{y}{h}$$

$$U^* = \frac{U}{U_0} \quad V^* = \frac{V}{U_0} \quad P^* = \frac{P}{\rho_0 U_0^2} \quad T^* = \frac{T - T_{min}}{T_{max} - T_{min}} \tag{4}$$

where $w$ and $h$ stand for experimental width and height respectively. Replace the dimensionless values back to Equation 3, yielding the normalized PDEs:

$$\frac{\partial U^*}{\partial x^*} + \frac{\partial V^*}{\partial y^*} = 0$$

$$U^*\frac{\partial U^*}{\partial x^*} + \gamma V^*\frac{\partial U^*}{\partial y^*} + \frac{\partial P^*}{\partial x^*} - \frac{1}{\text{Re}}\left(\frac{\partial^2 U^*}{\partial x^{*2}} + \gamma^2\frac{\partial^2 U^*}{\partial y^{*2}}\right) = 0$$

$$U^*\frac{\partial V^*}{\partial x^*} + \gamma V^*\frac{\partial V^*}{\partial y^*} + \gamma\frac{\partial P^*}{\partial y^*} - \frac{1}{\text{Re}}\left(\frac{\partial^2 V^*}{\partial x^{*2}} + \gamma^2\frac{\partial^2 V^*}{\partial y^{*2}}\right) + \text{Ri}\,T^* = 0 \tag{5}$$

$$U^*\frac{\partial T^*}{\partial x^*} + \gamma V^*\frac{\partial T^*}{\partial y^*} - \frac{1}{\text{Pe}}\left(\frac{\partial^2 T^*}{\partial x^{2*}} + \gamma^2\frac{\partial^2 T^*}{\partial y^{*2}}\right) = 0$$

and

$$\gamma = \frac{w}{h} \quad \text{Re} = \frac{U_0 w}{\nu} \quad \text{Pr} = \frac{\rho_0 c\nu}{k}$$

$$\text{Pe} = \text{Re} \cdot \text{Pr} \quad \text{Ri} = \frac{g\beta(T_{max} - T_{min})w}{U_0^2} \tag{6}$$

where dimensionless parameters $\gamma$, Re, Pr, Pe, and Ri represent aspect ratio, Reynolds number, Prandtl number, Peclet number, and Richardson number, respectively.

## A.2   ADDITIONAL MODEL DETAILS

**PINN Architecture**   We experimented two different types of PINN architecture, including 10 hidden layers with 100 neurons in each layer and 12 hidden layers with 200 neurons in each layer. No further experiments are conducted as the 10 layers one significantly under-predicted all field values, meaning it failed to capture the underlying physics pattern, The second one, which is being compared with PIGCN in this work, showed less accuracy while costing notably more time and graph memory to train.

**PIGCN Architecture**   As described in Section 4.4, we experimented various kinds of PIGCN architectures, which significantly impact performance of the model. Although deeper and

larger networks can improve the ability of the model to learn more complex features and relationships, over-complex model may lead to problems such as vanishing or exploding gradients and overfitting. In our experiments we first tested a uniform width model with 8 layers and 128 channels in each layer proposed by Peng et al. (2023), proving that our proposed architecture with symmetric number of hidden channels at $[16, 32, 64, 128, 64, 32, 16, 8]$ worked better. We also experimented symmetric architecture with different depth, including a shallow one with hidden channels of $[16, 32, 64, 32, 16, 8]$ and a deep one with hidden channels of $[16, 32, 64, 128, 256, 128, 64, 32, 16, 8]$, which suffer from under-emphasize and over-emphasize problems respectively.

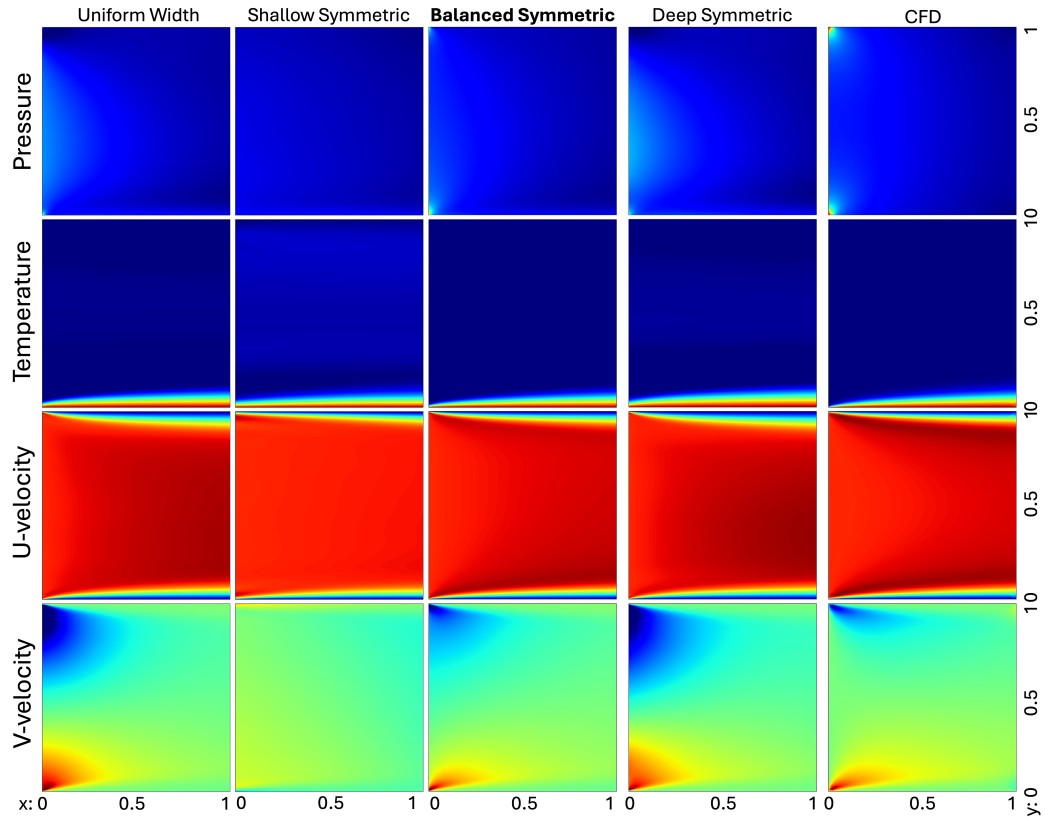

Figure 7: Comparison between PIGCN architectures.

### A.3 Additional Training Details

All models are trained with 100,000 iterations of Adam optimizer followed by L-BFGS optimzier until convergence on Nvidia A100 80GB GPU. Here convergence is defined by the overall loss of model not improving for 100 epochs. The learning rate for Adam optimizer is fixed at $5 \times 10^{-4}$ to ensure stable loss trend during training, and L-BFGS learning rate is set to 1 for faster convergence. When GradNorm is enabled, for both composite loss and PDE loss weighting, hyperparameter $\alpha$ is set to 2 which sets the strength of the restoring force. Weights are updated every 100 epochs to smooth the weighting trend and save computation time. All models uses sine function as activation function for its smoothness and differentiability. Algorithm 1 shows a general training pipeline of PIGCN.

### A.4 The Use of Large Language Models

This research does not involve LLMs as any important, original, or non-standard components. LLM was only used for editing and formatting purposes.

---

**Algorithm 1** PIGCN training pipeline

---

**Input**: Graph $\mathcal{G}$ constructed from mesh, boundary condition $B$, physical parameters $\mu$
**Output**: Fluid field predictions: $P, T, U, V$

1: Create boundary collocation nodes and attach to $\mathcal{G}$
2: Create interior collocation nodes and attach to $\mathcal{G}$
3: Initialize PIGCN model $M$: convolution layers $l$, sine activation function, set weights to $\mathbf{w}_0$
4: Initialize composite loss weights $\lambda(0) = (\lambda_{data}(0), \lambda_{bc}(0), \lambda_P(0), \lambda_{flux}(0))$ and weight update frequency $f_\lambda$     ▷ If training is data-free, $\lambda_{data} \leftarrow 0$ and will not be updated. If dynamic loss weighting is not enabled, $\lambda \leftarrow \lambda(0)$
5: Initialize PDE loss weights $\alpha(0) = (\alpha_c(0), \alpha_u(0), \alpha_v(0), \alpha_e(0))$ and weight update frequency $f_\alpha$     ▷ If dynamic PDE loss weighting is not enabled, $\alpha \leftarrow \alpha(0)$
6: Initialize Adam optimizer and max_adam_epoch
7: Initialize L-BFGS optimizer and patience
8: best_loss $\leftarrow \infty$
9: **for** $n \leftarrow 0$, max_adam_epoch **do**
10:     $\mathcal{P} = (P, T, U, V) \leftarrow M(G, \mathbf{w})$     ▷ Get prediction from model
11:     **if** is data-assisted training **then**
12:         Calculate data loss $L_{data}$
13:     **end if**
14:     Calculate boundary loss $L_{bc}$
15:     Calculate PDE residual $L_P = \alpha_c(n)L_c + \alpha_u(n)L_u + \alpha_v(n)L_v + \alpha_e(n)L_e$     ▷ Weighted PDE loss
16:     Calculate inlet flux $Q_{in} \leftarrow \int_{\Omega_{in}} \mathbf{u}_{in} \cdot \hat{\mathbf{n}}_{in} dL_{in}$
17:     Caluclate outlet flux $Q_{out} \leftarrow \int_{\Omega_{out}} \mathbf{u}_{out} \cdot \hat{\mathbf{n}}_{out} dL_{out}$
18:     $L_{flux} \leftarrow ||Q_{in} + Q_{out}||_2$
19:     $L \leftarrow \sum_i^{enabled} \lambda_i(n)L_i$     ▷ Weighted total loss
20:     **if** $L <$ best_loss **then** best_loss $\leftarrow L$
21:     **end if**
22:     **if** Dynamic composite loss weighting **and** $n \% f_\lambda = 0$ **then**
23:         $\lambda(n+1) = \text{GradNorm}(\lambda(n), L)$
24:     **end if**
25:     **if** Dynamic PDE loss weighting **and** $n \% f_\alpha = 0$ **then**
26:         $\alpha(n+1) = \text{GradNorm}(\alpha(n), L_P)$
27:     **end if**
28:     $\mathbf{w}_{n+1} \leftarrow \text{Adam}(\mathbf{w}_n, L)$     ▷ Update model weights
29: **end for**
30: patience_counter $\leftarrow 0$
31: **while** patience_counter $\leq$ patience **do**
32:     $L \leftarrow \sum_i^{enabled} \lambda_i(\text{max\_adam\_epoch})L_i$     ▷ Gather weighted loss terms
33:     **if** $L <$ best_loss **then** best_loss $\leftarrow L$
34:     **else** patience_counter $\leftarrow$ patience_counter $+1$
35:     **end if**
36: **end while**
37: Create evaluation graph $\mathcal{G}'$
38: $\mathcal{P} = (P, T, U, V) \leftarrow M(\mathcal{G}', \mathbf{w})$

---

