# OpenReview forum: "Physics-Informed Graph Convolutional Network for Data-Free Learning of Channel Flow Fields"
_ICLR.cc/2026/Conference — ICLR 2026 Conference Withdrawn Submission_

### Official Review · Reviewer_TnTr · 2025-10-27

**Soundness:** 3
**Presentation:** 3
**Contribution:** 3
**Rating:** 6
**Confidence:** 5

**Summary:**

The paper introduces PIGCN—a Physics-Informed Graph Convolutional Network—that replaces the dense fully connected layers of standard PINNs with graph convolutions to better capture local spatial coupling in fluid flows.
PIGCN is designed for data-free learning (training purely on PDE residuals + boundary conditions) and adds a two-level adaptive loss-weighting scheme plus a global flux-conservation loss to stabilize and accelerate training.

**Strengths:**

I love this paper because it shows it can work on fluid problem in this way. Graph architectures offer mesh-based inductive bias, naturally encoding locality and connectivity—ideal for fluid systems with irregular domains and boundary layers.

**Weaknesses:**

The study only validates PIGCN on steady, laminar, 2D channel flows with moderate Reynolds numbers.

Ablation study is not enough.

Justification is not enough (improving PINNs is not necessarily a good motivation).

Dynamic Loss Weighting is Empirical and Heuristic

The two-level GradNorm scheme introduces multiple hyperparameters (learning rates, normalization constants) that require manual tuning.

No theoretical analysis or convergence guarantee is provided for stability when loss gradients conflict.

**Questions:**

How small a GCN can be (still working)?

Can it be solved without Adam?

---

### Official Review · Reviewer_1qMu · 2025-10-30

**Soundness:** 1
**Presentation:** 1
**Contribution:** 1
**Rating:** 0
**Confidence:** 3

**Summary:**

This work investigates the use of GCN (Graph Convolutional Network) in PINN (Physics-Informed Neural Network), in addition to applying adaptive loss-weight balancing to facilitate effective model training. The authors compared the proposed model, PIGCN, with the vanilla PINN in straight and L-shaped channels, both in supervised and unsupervised settings.

**Strengths:**

1. The authors investigated both supervised and unsupervised settings, providing insights into the experimental performance of the models.

**Weaknesses:**

1. The motivation to focus on GCN is weak. The authors say “most PIGNN works were conducted on multi-layer perceptron (MLP)-based message-passing blocks, while only few preliminary works discussed the potential of graph convolutional network (GCN) despite its superior ability of learning local information by mixing a node feature with those of its neighbors,” but in terms of expressible power, GCN is weaker than the standard message-passing GNN. In addition, existing work[Gao+ Comput. Methods Appl. Mech. Eng. 2021] has already considered a PINN with GCN. Therefore, the authors should clarify the motivation and benefit of focusing on GCN rather than GNN.
2. The writing is not clear enough. The definition of $L_{flux}$ is missing. In addition, they provided almost no information regarding how the differentiation to compute the residual loss $L_P$ is performed. Since GCN is not a point-wise network and computational points interact, it is not obvious how to perform automatic differentiation with respect to spatial variables. The authors should clarify the definition of $L_{flux}$ and the method for computing differentiation in GCN.
3. Presentation of the results is difficult to follow. The authors should provide tables to summarize the results rather than writing numbers in the main text.
4. The authors might misunderstand the implementation of PINN. The vanilla PINN[Raissi+ J. Comput. Phys. 2019] uses the pointwise MLP, so GCN should be slower than that because the GCN model is pointwise MLP + graph convolution (it is, though unclear because no code is provided and a less informative description of the model construction). However, they say that “PIGCN required only around 60-70% of the PINN’s wall-clock time,” which seems to be a contradiction, provided that the number of layers and channels are comparable. In addition, the paper states, “PIGCN has 60% fewer trainable parameters than the dense network,” implying that the authors mistakenly implemented the vanilla PINN using a fully connected MLP that takes all computational points. Also, according to Table 1, PINN is reported to take more memory than PIGCN, which supports the reviewer’s insight. Even if the implementation is correct, minibatch training enables us to control the memory consumption of PINN. Therefore, the authors should clarify the correctness of the implementation of PINN and the reasons for the (seemingly) contradictory results.
5. The experimental evaluation is weak. The work uses only one baseline, vanilla PINN. At least, the authors should add PINNs with GNNs to demonstrate their practical superiority over GNNs. In addition, the authors used only one setting for PINN, whereas PIGCN uses various hyperparameters, making this a less fair comparison.
6. The practical and/or theoretical value of the research is unclear. For the unsupervised setting, the authors should compare with classical numerical solvers if they wish to claim practical benefit. In the supervised setting, the authors should also outline the practical or theoretical benefits of the proposed method.


Minor points:

* The subscript of $L_\mathrm{data}$ is in roman, but others are in italic. When it represents a word rather than the multiplication of multiple variables, the reviewer recommends using roman, e.g., $L_\mathrm{flux}$.
* p. 7: The reviewer cannot comprehend the meaning “=PINN”, maybe it is a typo?

**Questions:**

* The paper says, “While both models provided satisfactory prediction accuracy, data may not be available in many real-world cases, and it can limit generalization capability, which can require retraining of the model with any design changes,” but the reviewer cannot comprehend the meaning. Could the authors elaborate more on this point? Why does one condition require retraining and the other does not?
* According to Equation 4, x and y are nondimensionalized by different factors, which may distort the space and render the equation invalid. How do the authors guarantee that this scaling is correct?

---

### Official Review · Reviewer_WuFk · 2025-10-30

**Soundness:** 2
**Presentation:** 2
**Contribution:** 1
**Rating:** 2
**Confidence:** 5

**Summary:**

The authors enforce the governing physics law on the loss function of the graph neural network and solve the problem of flow in an L-shaped pipe. Visual comparison and error analysis have been shown to support the idea. The advantages of using graph neural networks over traditional MLPs in the PINNs framework are discussed. Computational time and GPU memory were considered as criteria for comparison between different methods.

**Strengths:**

1) Easy to read and understand
2) High-quality figures
3) Details of N.S. equations were presented.
4) Figure 1 is illustrative.

**Weaknesses:**

The combination of graph neural networks with PINNs is not novel and was presented for the first time 3 years ago (2022). Here is a link to the journal paper:

Physics-informed graph neural Galerkin networks: A unified framework for solving PDE-governed forward and inverse problems

https://doi.org/10.1016/j.cma.2021.114502

It is surprising that the authors referenced this article in their manuscript; however, they claimed that that was a data-driven method. Of course, they used sparse data for solving inverse problems. In the current manuscript, the authors also used data, as we can see in the schematic Figure 2 ($\frac{\partial L_{data}}{\partial w}$). I am surprised, as the title of the work is "PHYSICS-INFORMED GRAPH CONVOLUTIONAL NETWORK FOR DATA-FREE LEARNING OF CHANNEL FLOW FIELDS" and it contains the term "data-free". But in Figure 2 and Eq. 1 and Eq. 2, we see a component relevant to the data in the loss function.

For ICLR 2026, high-quality novel work is expected; given that this work is not new in terms of methodology, I cannot support it. The chosen test case is also not challenging and is even simpler than those solved in the above-mentioned journal paper. ICLR is a place for introducing novel frameworks and methodologies, not using a previously developed method to solve a new problem (or new application), which is not challenging.

--> As a side note, the abstract is too long.

--> There is no detail or even a high-level explanation of the graph neural network.

**Questions:**

Thanks for submitting this work to ICLR 2026. However, I am sorry to say that I do not have any specific questions, as this work does not present any novel ideas. Please see the Weaknesses box above. A suggestion: Try to use the framework for turbulent flow in 3D, which is indeed challenging.

---

### Official Review · Reviewer_TeTx · 2025-11-03

**Soundness:** 1
**Presentation:** 2
**Contribution:** 1
**Rating:** 2
**Confidence:** 4

**Summary:**

This authors propose a Physics-Informed Graph Convolution framework for solving 2D incompressible steady flow problems without observational/numerical simulation data. The proposed method uses graph convolutions to capture local spatial dependencies on irregular meshes. The authors also propose two-level dynamic loss weighting and a global flux-conservation loss to stabilize training and ensure global physical consistency. Experiments on rectangular and L-shaped channel flows show that PIGCN achieves better accuracy, faster training, and lower memory usage than PINNs.

**Strengths:**

1. The paper introduces a new Physics-Informed Graph Convolution method to address key limitations of traditional PINNs, such as poor scaling and difficulty in capturing localized interactions.

2. The proposed method shows significant improvements in both accuracy and memory efficiency compared to baseline PINN.

**Weaknesses:**

1. **Novelty claim**: The dynamic loss weighting has been proposed before in [5], albeit not on graph, but this undermines the novelty claim of the paper. I encourage the authors to articulate the novelty in contrast to [5] to better position the paper’s novelty.

2. **Lack of strong baselines**: One significant issue is that the paper does not discuss and benchmark against recently proposed strong GNN-CFD baselines such as MeshGraphNet [1], CFD-GCN [2], BSMS-GNN [4] and FVGCN [3].

3. **Lack of challenging benchmarks**: The datasets are quite simplistic in the sense  that objects of varying geometry in the flow field such as cylinder flow problem has not been considered. I urge the authors to consider such problems.

4. **Computational cost may not be realistic on large-scale problem**: Did you compute the derivative of the various losses with respect to model parameters w using auto-differentiation (refering to Figure 2)? That means you need to track gradients at each node of the graph including the boundary nodes. This could be computationally memory-heavy on large-scale mesh, which is often the case for CFD. This computational burden can be observed in Table 1 where PIGCN consumes 28 GB GPU memory when dealing with only 144K nodes. This is not practical on large-scale CFD problems with large-scale mesh.

5. **Lack of clarity**: How did you discretize the computation of $L_c, L_u, L_v, L_e$ given that the input is a graph?

6. **Lack of discussion about limitations**: To my understanding, the method  has the limitation that it needs to be retrained when the flow conditions remain same except only the geometry of the object changes. Can you confirm if that is the case?

7. **Stability of Loss coefficient**: Can you discuss and demonstrate the stability of the eight loss coefficients in Equation 2?

8. **Figure ambiguity**: The stack of graphs in GCN block in Figure 2 is confusing. Please consider revising the figure.

[1]  Learning Mesh-Based Simulation with Graph Networks, ICLR 2021

[2] Combining Differentiable PDE Solvers and Graph Neural Networks for Fluid Flow Prediction, ICML 2020.

[3] Finite Volume Features, Global Geometry Representations, and Residual Training for Deep Learning‑based CFD Simulation, ICML 2024.

[4] Efficient Learning of Mesh-Based Physical Simulation with Bi-Stride Multi-Scale Graph Neural Network, ICML 2023.

[5] Dynamic Weight Strategy of Physics-Informed Neural Networks for the 2D Navier–Stokes Equations, Entropy 2022.

**Questions:**

Rather than computing global flux conservation, was there a reason not to consider computing averaged local flux conservation at each cell, as done in finite volume? + Weaknesses 1, 4-7.

---

### Note · Authors · 2025-12-02

I have read and agree with the venue's withdrawal policy on behalf of myself and my co-authors.